# Development of loop-mediated isothermal amplification (LAMP) assay for rapid and direct screening of yellowfin tuna (*Thunnus albacares*) in commercial fish products

**Ashraf Ali**[1], **Antonia Kreitlow**[2], **Madeleine Plötz**[2], **Giovanni Normanno**[1], **Amir Abdulmawjood**[2]*

**1** Department of Sciences of Agriculture, Food, Natural Resources and Engineering (DAFNE) University of Foggia, Foggia, Italy, **2** Institute of Food Quality and Food Safety, University of Veterinary Medicine Hannover, Hannover, Germany

* amir.abdulmawjood@tiho-hannover.de

## Abstract

Tuna is one of the most widely consumed fish on the European market, being available in various consumable options. Among them, *Thunnus albacares*, also called yellowfin tuna, is a delicacy and is consumed by millions of people around the world. Due to its comparatively high cost and demand, it is more vulnerable to fraud, where low-cost tuna or other fish varieties might be replaced for economic gain. In this study, a loop-mediated isothermal amplification (LAMP) assay was developed and validated for targeting the mitochondrial *cytochrome b* gene for fast and direct detection of *Thunnus albacares*, which is a valuable tuna species. The analytical specificity was confirmed using 18 target samples (*Thunnus albacares*) and 18 samples of non-target fish species. The analytical sensitivity of the LAMP assay was 540 fg DNA per reaction. In addition, a simple and direct swab method without time-consuming nucleic acid extraction procedures and the necessity for cost-intensive laboratory equipment was performed that allowed LAMP detection of *Thunnus albacares* samples within 13 minutes. Due to its high specificity and sensitivity, the LAMP assay can be used as a rapid and on-site screening method for identifying *Thunnus albacares*, potentially providing a valuable monitoring tool for food authenticity control by the authorities.

## Introduction

Tuna belongs to the *Thunnus* tribe (scombridae) family, and it is one of the most popular delicacies among marine fish globally. Tuna is a very large species which is widely distributed in oceanic environments. It is consumed in various forms, either fresh or as canned products. The Council Regulation (EEC) No. 1536/92 defines relevant members of the genus *Thunnus* on the European market, which include yellowfin tuna (*Thunnus* (*neothunnus*) *albacares*) and skipjack tuna (*Euthynnus* (*Katsuwonus*) *pelamis*). These two species dominate the market share with more than 80%, with yellowfin tuna belonging to the higher price category [1].

**Funding:** The author(s) received no specific funding for this work.

**Competing interests:** The authors have declared that no competing interests exist.

Tuna is considered as a healthy source of animal proteins, "omega-3" fatty acids and bioactive nitrogen compounds, which make it more popular among many fish species. Due to its bulky size, it is not possible to sell tuna as one piece so that it is usually traded in portions or types of fillet. Alternatively, it is offered as a floss product after being dried, flavoured, roasted and rolled. Tuna is mainly used for canning and making sashimi and sushi products. The characteristic texture, taste and flavour of canned tuna products, available either in oil or brine, makes it a delicacy for consumers. Among the designated tuna species, yellowfin tuna (*Thunnus albacares*) is one of the most widely used species in canned products.

Prices for different tuna species vary according to species and popularity. The prices are regulated by species of tuna, trade performance and ultimate use. The market price of raw tuna, mostly used in sashimi and sushi, is quite high. Sashimi preparation requires high quality fish and usually is the most demanding and expensive option on the market [2]. Compared to fresh and raw tuna, quality requirements are not very stringent for canned, cooked, frozen and smoked tuna products. Prices for skipjack tuna are much lower than for bigeye, bluefin and yellowfin tuna due to market-specific influences and consumption patterns in European countries [1].

There are studies which indicate mislabelling of various tuna products, particularly skipjack and yellowfin tuna [2, 3]. Due to price variations and taxes imposed by the European Union on imported products, attention has been given to methods which can identify canned fish species in accordance with EU regulations. Several methods based on protein and DNA detection are available to differentiate between fish species. For example, raw tuna and other fish products can be identified using the isoelectric focusing (IEF) method, which is based on the availability of water-soluble proteins in the fish species [4]. Proteomics-based approaches like peptide mass fingerprint (MALDI-TOF) and peptide fragment fingerprinting (MALDI-LIFT TOF/TOF) separated by 1D SDS-PAGE have also been applied for tuna species identification [5]. These methods are based on the analysis of proteins present in by-products of canned tuna as well as in products obtained from filleting the skeletal red muscle of fresh fish [5]. Apart from the high labour input involved in performing this procedure, steps like canning, cooking and smoking involve intense heat treatment which can cause irreversible loss of water and affects solubility of proteins, hampering the identification process [6]. Methods based on antibody-antigen interaction can provide suitable alternatives, but until now, only a handful of immunoassays have been developed, and none of them are available for large-scale application for routine analysis. DNA amplification-based methods such as PCR–RFLP [7], PCR-SSCP [8] or real-time PCR [9] as well as sequencing-based methods are also applicable for fish species detection [10]. However, performing PCR requires time-consuming cycling steps, costly and bulky equipment as well as a laboratory environment, thus limiting its utility for on-site or rapid diagnosis.

There are several isothermal amplification methods that can amplify DNA at a constant temperature using a simple water bath or heating block without the need for expensive thermal cyclers and detection systems. The isothermal methods include Strand Displacement Amplification (SDA), Rolling Circle Amplification (RCA), the Ramification Amplification Method (RAM), Isothermal Multiple Displacement Amplification (IMDA), Nucleic Acid Sequence Based Amplification (NASBA), Helicase-Dependent Amplification (HAD), Recombinase Polymerase Amplification (RPA), Polymerase Spiral Reaction (PSR) methods and Loop-Mediated Isothermal Amplification (LAMP). The above have been applied for detecting several infectious and non-infectious diseases of animals and humans. Each method has its advantages and disadvantages [11]. In the past few years, the nucleic acid amplification technique called loop-mediated isothermal amplification (LAMP) has gained increasing popularity. This method can be used for fast identification of different targets, generally offering possibilities

for on-site application. LAMP uses four to six specifically designed primers that recognise six to eight regions on the target gene sequence, resulting in high sensitivity and specificity. Additionally, the complete identification process can be performed under isothermal conditions in less than 30 minutes. In the last few years, many studies have been performed using LAMP for food authentication, including eel [12], salmon [13], cod [14], tuna [15] and ostrich meat [16].

The aim of the present study was to develop a new LAMP assay for direct detection of yellowfin tuna (*Thunnus albacares*) in commercial food products, targeting the mitochondrial cytochrome *b* gene (*cytB*).

## Material and methods

### Sample collection

A total of 36 samples were collected from different supermarkets in Hannover, Germany. Of these, 18 samples were labelled as *Thunnus albacares* steaks. The remaining 18 non-target samples included six canned samples of *Katsuwonus pelamis*, two *Oncorhynchus keta* fillets, one *Onchorhynchus nerka* fillet, one *Salmon salar* fillet, five *Gadus morhua* fillets and three *Gadus chalcogrammus* fillets. The samples were stored in accordance with the manufacturers' specifications at -20°C or at room temperature until use. Manufacturer information is available as supplementary data (Table 1s in S1 Data).

### DNA extraction

DNA extraction was performed using the DNeasy Blood &Tissue Kit (Qiagen GmbH, Hilden, Germany) in accordance with the manufacturer's instructions. The concentration and purity of extracted DNA was checked using the spectrophotometer Nanodrop 2000c (Thermo Fisher scientific, VWR International GmbH, Darmstadt, Germany). Extracted DNA was stored at 4°C until use.

### Designing of species-specific LAMP primers

The mitochondrial cytochrome *b* (*cytB*) gene sequence (accession no. JN086153.1) of *Thunnus albacares* was retrieved from GenBank (https://www.ncbi.nlm.nih.gov/genbank/) and used as a target for species-specific LAMP primer design. In a first step, the template sequence was aligned with several *cytB* gene sequences of the target species obtained from the National Centre for Biotechnology Information (NCBI) (Bethesda, MD, USA). The alignments were performed using the Basic Local Alignment Search Tool (BLAST) by NCBI and are available as supplementary material (Figure 1s in S1 Data). Homologous regions with 100% coverage of nucleotide bases were selected as potential primer-binding sites. Alignment of *cytB* genes was also performed with relevant non-target species on the European market like skipjack tuna and other fish species to ensure that cross-reactions would be avoided by sufficient mismatches within the primer binding sites.

Six oligonucleotide primers, including forward and backward outer primers (F3 and B3), forward and backward inner primers (FIP and BIP) as well as forward and backward loop primers (LF and LB), were designed using the LAMP Designer software, ver. 1.10 (PREMIER Biosoft, CA, USA) and synthesised in HPSF purified quality by Eurofins Genomics GmbH (Ebersberg, Germany). Primer sequences are shown in Table 1.

### Analytical specificity and sensitivity of the *Thunnus albacares* LAMP assay

For determining analytical specificity of the newly established *cytB* LAMP assay, 18 *Thunnus albacares* samples and 18 samples of other fish species were comparatively tested with real-

**Table 1.** *cytB* gene-targeting LAMP primer sequences.

| Designation | Primer Sequence | Length | Position (NCBI acc. no. JN086153.1) |
|---|---|---|---|
| Tuna F3 | ATACGCAATTCTTCGGTCC | 19 bp | 15212–15230 |
| Tuna B3 | TTGTTCTCAGCTCAGCCT | 18 bp | 15488–15505 |
| Tuna LF | GAAGTGTGCAGGAAGGGAA | 19 bp | 15292–15310 |
| Tuna LB | GCGGAACAGCCCTTCATTA | 19 bp | 15408–15426 |
| Tuna FIP | TGGTCGGAATGTTAGAGTT–CGCAGCCTCCATCCTTGTACTT | 41 bp | 15317–15338 |
| | | | 15266–15284 |
| Tuna BIP | TGCAGACGTAGCCATTCTT–ACCAGGCTACTTGGCCGATAA | 40 bp | 15368–15389 |
| | | | 15427–15444 |

time PCR. Analytical sensitivity of LAMP assay was defined by testing 10-fold serial diluted DNA of *Thunnus albacares* using AE buffer (Qiagen Ltd.) as a dilution medium. Dilutions contained 54 ng/µL to 0.54 pg/µL of DNA. Each dilution was tested in triplicate.

## Direct detection by MSwab

For direct detection of *Thunnus albacares* without elaborate sample processing steps, a swab method was performed using the MSwab kit (Copan S.p.A., Brescia, Italy) [17]. The dry swab was picked from the supplier packet and scrubbed and rolled firmly several times over the surface of two steak samples designated as *Thunnus albacares*. The swabs were then dipped into the MSwab® buffer tubes with the swab head detached and remaining inside. Tubes were shaken vigorously by hand without any vortexing or incubation. After proper mixing, 5 µL from each buffer tube were directly used as a template for LAMP testing. Besides the regularly run positive and negative control reactions, one kit-extracted sample of *Salmon salar* (Sal 1) was taken as negative extraction control during LAMP testing of the swab samples.

## LAMP assay

LAMP reactions were performed using the portable real-time fluorimeter Genie II® (Optigene Ltd., Horsham, UK). Each reaction mixture with a total volume of 25 µL consisted of 15 µL GspSSD Isothermal Mastermix (ISO-001) (OptiGene Ltd.), 2.5 µL primer mix, 2.5 µL Nuclease-free water and 5 µL template DNA. The primer mix was prepared in a standard version in accordance with the recommendations of OptiGene Ltd. (http://www.optigene.co.uk/support/). In each reaction, the concentration of F3 and B3 primers was 0.2 µM, of FIP and BIP primers 0.8 µM and of LF and LB primers 0.4 µM. The LAMP reaction was performed at 65˚C for 3040 minutes followed by melting curve analysis for descending temperatures between 98˚C–80˚C (ramp rate: 0.05˚C/s). In each run, a positive and a negative control reaction were performed using 5 µL of DNA template (0.1 ng/µL) from *Thunnus albacares* (T4) and 5 µL of nuclease-free water instead of the DNA template, respectively.

## PCR assay

A slightly modified real-time PCR assay based on the descriptions of Lee et al. [18] was performed to confirm the analytical specificity of the LAMP assay. Each reaction mixture with a total volume of 30 µL contained 15 µL FastStart Essential DNA Green Master (Roche Diagnostics GmbH, Mannheim, Germany), 1 µL of Alba-F and Alba-B primers (10 µM) each, 10 µL PCR gradient water and 3 µL DNA template. The thermocycling process was started with a preincubation step for 10 min at 95˚C, followed by 35 three-step amplification cycles, each including 30-s periods of denaturation at 95˚C, annealing at 64˚C and extension at 72˚C.

Subsequently, a final extension step was performed at 72°C for 5 min. Amplification was followed by melting curve generation at 95°C for 10 s, 65°C for 60 s and 97°C for 1 s. All PCR reactions were carried out using the real-time LightCycler96 system (Roche Diagnostics GmbH). Primers were ordered in HPSF purified quality (Eurofins Genomics GmbH).

## Gel electrophoresis

In addition to evaluating the melting temperatures, the specificity of the *Thunnus albacares* LAMP products obtained from 10-fold serial dilutions was additionally confirmed using gel agarose electrophoresis. For this purpose, a 2% gel was prepared with agarose powder (Universal, VWR International GmbH, Darmstadt, Germany) and GelRed™ (Biotium, Eching, Germany) as staining agent. Gel electrophoresis was performed at 5 volts/cm for 90 minutes to observe the characteristic ladder-like pattern of the LAMP products. After electrophoresis, the DNA was visualised and documented with gel doc (Gel Doc EZ Imager, Bio-Rad Laboratories GmbH, Munich, Germany). The obtained results were compared with DNA ladders of a marker (Biozyme) including DNA band sizes from 2000 bp to 50 bp (Quantitas, Biozym Scientific GmbH, Hessisch Oldendorf, Germany).

## Results

### Analytical specificity and sensitivity of the LAMP assay

The specificity of the LAMP assay was tested using 18 samples of the target species *Thunnus albacares* and 18 fish samples from five different non-target species (Table 2). All *Thunnus albacares* samples were amplified, while no amplification occurred when DNA templates from the 18 other fish species were used. Melting curve analysis by Genie II® showed specific melting temperatures of 85.4 ± 0.8°C for all *Thunnus albacares* samples (Fig 1). The results of LAMP and PCR were fully compliant.

The analytical sensitivity of the *cytB* LAMP assay was determined on the basis of 10-fold serial DNA dilutions. DNA amounts of 54 ng/μL to 0.54 pg/μL were successfully detectable in all three repetitions. Thus, the detection limit observed for *Thunnus albacares* was 0.54 pg/μL (Table 3, Fig 2).

### Direct detection using MSwab

To avoid time-consuming DNA extraction, a fast and simple sample preparation method using the MSwab kit was performed. Amplification was observed in all swab samples from *Thunnus albacares* with detection times between 13 and 17 minutes (Fig 3) and specific melting temperatures of 85.4 ± 0.8°C.

### Detection by gel electrophoresis

Gel electrophoresis was performed with LAMP products obtained after amplification of the 10-fold serially diluted tuna reference DNA samples used for determining analytical sensitivity. Identical ladder-like DNA bands of all LAMP products were clearly visible after the run, indicating that all amplicons were specifically amplified (Fig 4).

## Discussion

Proper identification of fish species is very important for public health as well as for prevention of food fraud. The health risks associated with fish fraud are exposure to allergens, consumption of poisonous fish, accumulation of toxic metals, diarrhoea and sometimes death [19, 20]. Tuna consumption sometimes causes "scombroid poisoning" [20]. Such incidents indicate

**Table 2. Analytical specificity of the *cytB*-LAMP assay for target and non-target species compared to real-time PCR.**

| Sample no. | Sample ID | Species | LAMP Amplification (mm:ss) | LAMP Melting (C°) | Real-time PCR |
|---|---|---|---|---|---|
| 1 | T3 | *Thunnus albacares* | 7:45 | 85.5 | + |
| 2 | T4 | *Thunnus albacares* | 7:00 | 85.4 | + |
| 3 | T5 | *Thunnus albacares* | 7:45 | 85.5 | + |
| 4 | T6 | *Thunnus albacares* | 7:30 | 85.3 | + |
| 5 | T7 | *Thunnus albacares* | 7:15 | 85.4 | + |
| 6 | T8 | *Thunnus albacares* | 7:30 | 85.3 | + |
| 7 | T25 | *Thunnus albacares* | 8:45 | 85.4 | + |
| 8 | T26 | *Thunnus albacares* | 9:00 | 85.5 | + |
| 9 | T27 | *Thunnus albacares* | 9:30 | 85.5 | + |
| 10 | T28 | *Thunnus albacares* | 9:30 | 85.5 | + |
| 11 | T29 | *Thunnus albacares* | 10:00 | 85.1 | + |
| 12 | T30 | *Thunnus albacares* | 9:30 | 85.4 | + |
| 13 | T31 | *Thunnus albacares* | 9:30 | 85.4 | + |
| 14 | T32 | *Thunnus albacares* | 9:15 | 85.4 | + |
| 15 | T34 | *Thunnus albacares* | 10:00 | 85.3 | + |
| 16 | T35 | *Thunnus albacares* | 10:00 | 85.3 | + |
| 17 | T37 | *Thunnus albacares* | 10:00 | 85.4 | + |
| 18 | T38 | *Thunnus albacares* | 9:45 | 85.4 | + |
| 19 | ANT1 | *Oncorhynchus keta* | – | – | – |
| 20 | ANT2 | *Oncorhynchus keta* | – | – | – |
| 21 | ANT3 | *Salmon salar* | – | – | – |
| 22 | ANT4 | *Oncorhynchus nerka* | – | – | – |
| 23 | ANT5 | *Gadus morhua* | – | – | – |
| 24 | ANT6 | *Gadus morhua* | – | – | – |
| 25 | ANT7 | *Gadus morhua* | – | – | – |
| 26 | ANT8 | *Gadus morhua* | – | – | – |
| 27 | ANT9 | *Gadus morhua* | – | – | – |
| 28 | ANT10 | *Gadus chalcogrammus* | – | – | – |
| 29 | ANT11 | *Gadus chalcogrammus* | – | – | – |
| 30 | ANT12 | *Gadus chalcogrammus* | – | – | – |
| 31 | T15 | *Katsuwonus pelamis* | – | – | – |
| 32 | T16 | *Katsuwonus pelamis* | – | – | – |
| 33 | T17 | *Katsuwonus pelamis* | – | – | – |
| 34 | T18 | *Katsuwonus pelamis* | – | – | – |
| 35 | T20 | *Katsuwonus pelamis* | – | – | – |
| 36 | T23 | *Katsuwonus pelamis* | – | – | – |

severe risks associated with mislabelling and species substitution of fish which also hampers consumer trust in the market.

Due to growing economic value, species substitution in food products is widespread, leading to health risks and economic losses. The European regulation (EC) 178/2002 requires full traceability of a product throughout the food chain and provides a legal basis for combatting food fraud. In accordance with the European regulation (EC) 1169/2011, all types of meat products must be correctly labelled with proper nomenclature. More specifically, in accordance with EU regulation 1379/2013, seafood labelling requires the inclusion of commercial and scientific names and information about whether the item was deforested [21, 22].

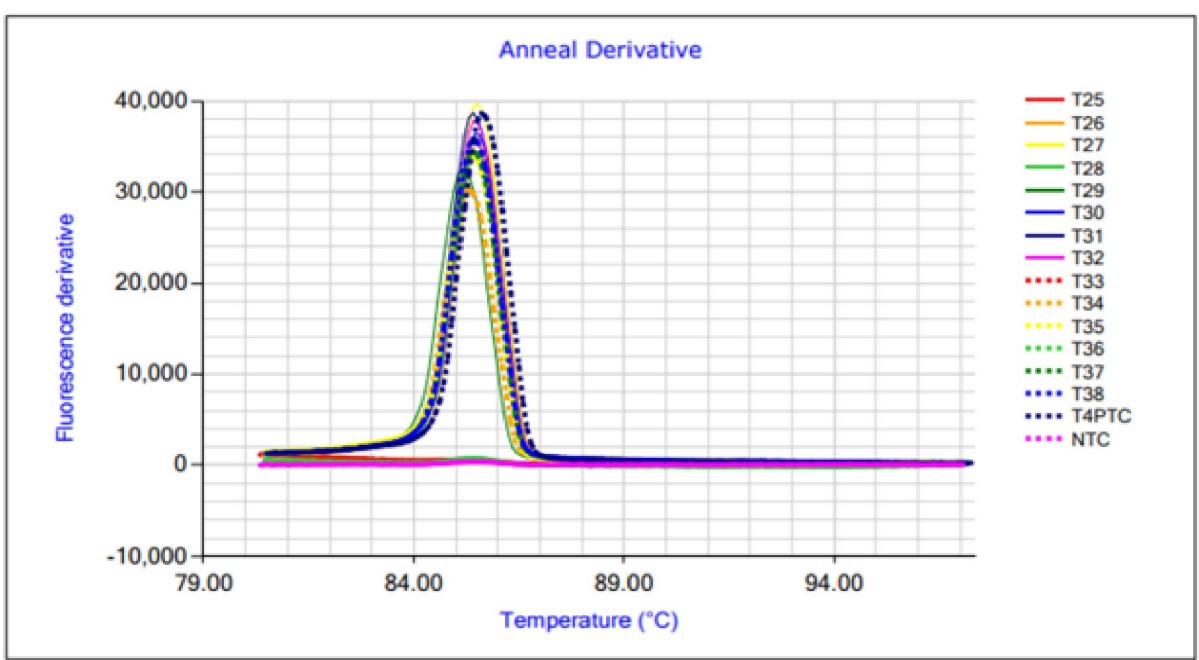

**Fig 1. Melting curves of LAMP products from various *Thunnus albacares* DNA samples.** The assay showed specific melting temperatures of 85.4 ± 0.8˚C. T4PTC = positive control; NTC = non-template control.

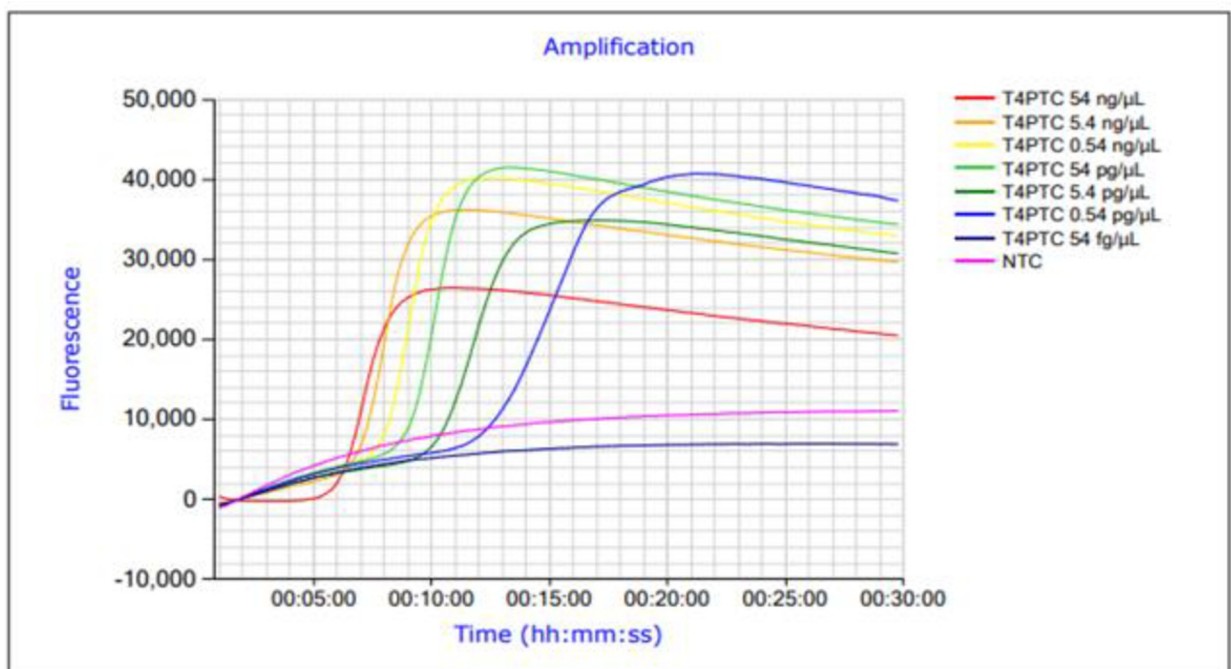

**Fig 2. Amplification profile of the *Thunnus albacares* LAMP assay using different dilutions.** The samples contained DNA concentrations of 54ng/μL to 54 fg/μL. NTC = non-template control.

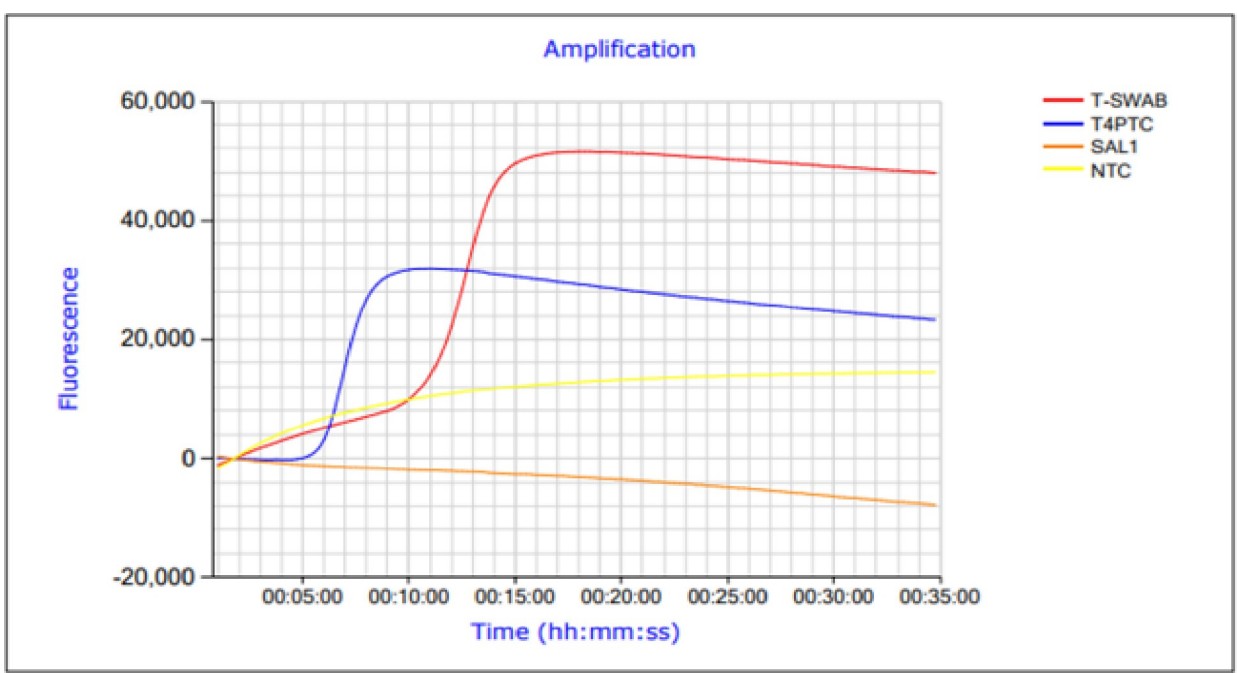

**Fig 3. Amplification profile of the *Thunnus albacares* LAMP assay using MSwab extraction of DNA.** T-SWAB = DNA from *Thunnus albacares* extracted by MSwab method; T4PTC = positive control; SAL1 = DNA from *Salmon salar* (negative extraction control); NTC = non-template control.

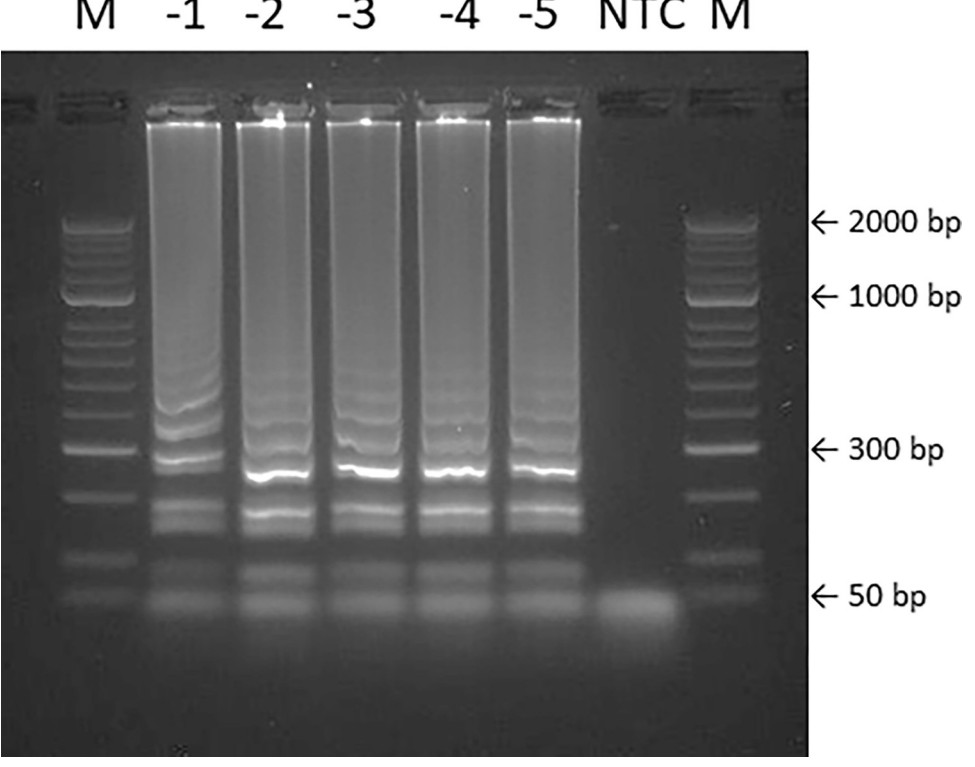

**Fig 4. Confirmation of the specificity of LAMP products obtained from 10-fold serially diluted tuna DNA samples using gel electrophoresis.** All samples showed an identical ladder-like band profile.

**Table 3. Analytical sensitivity of the LAMP assay using serial dilutions of *Thunnus albacares* DNA.**

| Dilution | $10^{-0}$ | $10^{-1}$ | $10^{-2}$ | $10^{-3}$ | $10^{-5}$ | $10^{-5}$ | $10^{-6}$ |
|---|---|---|---|---|---|---|---|
| Total DNA amount | (54 ng/µL) | (5.4 ng/µL) | (0.54 ng/µL) | (54 pg/µL) | (5.4 pg/µL) | (0.54 pg/µL) | (54 fg/µL) |
| Amplification time | | | | | | | |
| 1st run | 7:15 | 8:00 | 9:00 | 10:15 | 11:45 | 14:30 | - |
| 2nd run | 7:30 | 8:15 | 9:30 | 10.45 | 12:15 | 13:15 | - |
| 3rd run | 6:45 | 7:30 | 8:45 | 9:45 | 11:45 | 13:45 | - |
| Mean | 6.97 | 7.82 | 8.92 | 10.1 | 11.68 | 13.63 | - |
| SD± | 0.45 | 0.45 | 0.43 | 0.51 | 0.41 | 0.60 | - |

There are several molecular methods available for fish species detection. Among them, LAMP has gained in popularity in recent decades and is one of the most widely researched methods for species identification [11, 23]. Thus, it has already been used to detect several fish species such as eel (*Anguilla anguilla*) [12], salmon (*Oncorhynchus mykiss*) [24], cod (*Gadus morhua*) [14] and skipjack tuna (*Katsuwonus pelamis*) [15]. LAMP was applied successfully in various contexts and appears to be a fast and efficient alternative to PCR in the case of food fraud monitoring due to its low equipment requirements, easy handling and ability to be used in field [11]. The present study describes a LAMP assay for detecting *Thunnus albacares*. Even though other methods are available for tuna species detection, such as an investigation by PCR-RFLP [25], species-specific conventional PCR or real-time PCR [17], those are time-consuming and elaborate. In contrast, the *cytB* gene-based LAMP assay provides a robust, sensitive and specific tool for fast on-site detection of *Thunnus albacares* in less than 15 minutes.

The most challenging task in establishing a LAMP assay is the design of species-specific primers [14]. Both nuclear DNA and mitochondrial DNA were successfully used for primer design in fish species identification [14, 26, 27]. The most targeted mitochondrial genes for species identification include *cytB*, 12S rRNA, 16S rRNA and the cytochrome c oxidase subunit I (COX1) [28]. The *cytB* gene proved to be a successful marker for identifying seafood species and resolving species phylogenies [29]. Recently, Kappel et al. (2017) demonstrated the feasibility of an Illumina MiSeq NGS method targeting two short fragments of the mitochondrial *cytB* gene to authenticate tuna samples containing mixtures of species [30]. Another study in 2017 reported that the *cytB* mini-barcode was more competent than COI in the distinction of four Gadidae species (*G. morhua*, *G. macrocephalus*, *Theragra chalcogramma*, *Pollachius virens*) using HRM analysis [31]. There are other studies where *cytB* was successfully used for tuna species detection [32–34]. It was also a preferred target region for the development of LAMP assays detecting several fish and other animal species [12, 15, 16].

In this study, all *Thunnus albacares* samples collected from different supermarkets were correctly amplified using the newly established *cytB* LAMP assay. None of the other fish species used for control purposes displayed any cross-reaction. The high specificity of the LAMP assay for the authentication of *Thunnus albacares* was ensured by suitable LAMP primers that covered sufficient mismatches within the corresponding target regions of other fish species [14]. Due to a high degree of homology between the *cytB* sequences of the closely related tuna species *Thunnus alalunga*, *Thunnus thynnus* and *Thunnus obesus*, cross-reactions with the current LAMP primers might occur. However, this was not tested for in the present study, as the effect can be neglected due to the low European market dominance of these tuna species. In local supermarkets, generally only yellowfin and skipjack tuna products are available.

Other LAMP studies on detection of fish species showed varying degrees of analytical sensitivity. Very sensitive assays were able to detect up to 5 fg of skipjack tuna [35]. In another LAMP assay, 50 pg of skipjack tuna was detectable [15]. Further LAMP assays detected up to

500 pg of eel (genus *Anguilla*) [12], 285 pg of Atlantic cod (*Gadus Morhua*) [36], 37 pg of Pacific cod (*Gadus macrocephalus*) [36] and 197 pg of haddock (*Melanogrammus aeglefinus*) [36]. Compared to these results, the LAMP assay described in this study showed a high analytical sensitivity of 0.54 pg/μL. It enabled one of the main advantages of the established *cytB* LAMP assay, namely the simple and direct detection of *Thunnus albacares* in steak & fillet products without using any kit-based DNA extraction process. While kit extraction requires at least one time-consuming incubation step for accurate sample lysis as well as several washing steps and elution, dabbing samples with MSwab allowed direct detection of the target species with only marginally longer detection times compared to the kit-extracted DNA (Fig 3).

The LAMP assay was robust against the presence of PCR inhibitor substances like salt, spices and oil as it was able to detect DNA from various processed and canned products [37]. The short amplification time and low susceptibility towards inhibitors compared with PCR are noteworthy properties, enabling direct and reliable investigation of food samples in restaurants and of canned products [37, 38].

Overall, the *cytB* LAMP assay provided an accurate identification of the target species within 20 minutes using portable equipment. The application of the newly established *cytB* LAMP in food analysis will enable carrying out cost-effective and easy-to-perform tests in the field.

## Conclusion

The LAMP assay described in this study provides a rapid, easy-to-perform, cost-effective and reliable method for identifying *Thunnus albacares* in tuna steaks, fillets or canned products. Highly specific primers based on the *cytB* gene allow for selective detection of the target species, avoiding cross-reaction with relevant non-target types of fish on the European market. Without the necessity for a complex DNA extraction process, the *cytB* LAMP assay can directly identify *Thunnus albacares*-containing products in less than 20 minutes. It may significantly help in detecting mislabelled products and cases of food fraud, and could therefore potentially be used by food monitoring authorities in retail, import trade surveillance, processing plants, restaurants or communal catering facilities.

## Supporting information

**S1 Data.**
(PDF)

**S1 Raw images.**
(TIF)

## Acknowledgments

The authors would like to thank Anke Bertling for her excellent technical assistance.

## Author Contributions

**Conceptualization:** Ashraf Ali, Antonia Kreitlow, Madeleine Plötz, Giovanni Normanno, Amir Abdulmawjood.

**Data curation:** Antonia Kreitlow.

**Formal analysis:** Ashraf Ali, Giovanni Normanno, Amir Abdulmawjood.

**Funding acquisition:** Amir Abdulmawjood.

**Investigation:** Ashraf Ali, Giovanni Normanno, Amir Abdulmawjood.

**Methodology:** Ashraf Ali.

**Resources:** Madeleine Plötz.

**Software:** Amir Abdulmawjood.

**Supervision:** Amir Abdulmawjood.

**Validation:** Ashraf Ali, Amir Abdulmawjood.

**Visualization:** Antonia Kreitlow.

**Writing – original draft:** Ashraf Ali.

**Writing – review & editing:** Antonia Kreitlow, Madeleine Plötz, Giovanni Normanno, Amir Abdulmawjood.

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
