## [Decision Letter · Decision Letter 0]

23 May 2022

PONE-D-22-09199Development of loop-mediated isothermal amplification (LAMP) assay for rapid and direct screening of yellowfin tuna (Thunnus albacares) in commercial fish productsPLOS ONE

Dear Dr. Abdulmawjood,

Thank you for submitting your manuscript to PLOS ONE. After careful consideration, we feel that it has merit but does not fully meet PLOS ONE’s publication criteria as it currently stands. Therefore, we invite you to submit a revised version of the manuscript that addresses the points raised during the review process.

The subject of the present submission is interesting and up to date. However, some issues raised during the peer review process need to be carefully considered by the authors and the paper should be thoroughly revised before it can be considered for publication. One of the main issues is that regarding the specificity of the applied method. The specificity of the software-designed primers needs to be cross-checked with closely related tuna species. Moreover, the authors are required to check the English of the manuscript and to consider all the reviewer’s comments during the manuscript revision.

We look forward to receiving your revised manuscript.

Kind regards,

Aldo Corriero, Ph.D.

Academic Editor

PLOS ONE

Journal Requirements:

2. Please include information in your Methods section on the location and vendor of the products analysed, in enough detail fro another researcher to reproduce the findings.

Reviewers' comments:

Reviewer's Responses to Questions

**Comments to the Author**

1. Is the manuscript technically sound, and do the data support the conclusions?

Reviewer #1: Partly

Reviewer #2: Yes

Reviewer #3: Partly

2. Has the statistical analysis been performed appropriately and rigorously? 

Reviewer #1: No

Reviewer #2: Yes

Reviewer #3: No

3. Have the authors made all data underlying the findings in their manuscript fully available?

Reviewer #1: No

Reviewer #2: No

Reviewer #3: No

4. Is the manuscript presented in an intelligible fashion and written in standard English?

Reviewer #1: Yes

Reviewer #2: No

Reviewer #3: No

5. Review Comments to the Author

Reviewer #1: The manuscript is about the development of loop-mediated isothermal amplification (LAMP) assay for a fast and reliable identification of yellowfin tuna (Thunnus albacares) in commercial fish products. The topic is consistent with the scope of the journal and the manuscript is generally well structured. However, there are so many inaccuracies in the text and methodology, several changes in typos and spelling are needed and some information appear incomplete.

The text appears descriptive but lacks appropriate genetic studies and analysis.

For all these reasons, in my opinion, the paper should be thoroughly revised and is not suitable for publication in its current form.

Specific suggestions

Line 42: the Council Regulation (EEC) No 1536/92 defines “Tuna” all the species belonging to the genus Thunnus, please, better specify this concept.

Line 42: did you mean Thunnus thynnus? Please replace Thunnus thunus

Line 42: replace “Bluefin tuna” with “bluefin tuna”

Line 43: please, replace Thunnus alalanga with Thunnus alalunga

Line 45: please replace “omega 3 fatty acid” with “omega-3 fatty acid”

Line 54: please remove “which is”

Line 58: please replace “big eye tuna” with “bigeye tuna”

Line 58: please, cite also the bluefin tuna, that currently is the most expensive species

Line 60: some sentence about frauds for tuna species substitution, and some related reference, needs to be added. Further, about “variation in prices and tax imposed by the European Union on imported products” add some reference or information about imported products numbers

Line 63: also, proteomics analysis can be applied for tuna species identification in raw products, please cite them and add references

Line 70: please try to change the sentence by avoiding the use of the word “which” too many times

Line 74: The sentence “need longer time to pursue” give an incomplete information, since using specie-specific primers timing is very short. Thus, could be better to specify this concept and highlight the needs to design specie-specific reactions to shorten time analysis

Line 80: please, replace “cytochrome b” with “cytochrome-b”

Line 82: please, replace “food” with “fish”, since you only cited papers related to this food category

Line 90: specify how many steaks/canned samples you evaluated

Line 93: please replace “done” with “performed”.

Line 101: please replace “cytochrome b” with “cytochrome-b”.

Line 104: please, better specify what do you mean when you speak about “several target species”. The target species should be one (Thunnus albacares). Thus, do you mean several sequences of different specimens of the target species obtained from NCBI? Please, better explain the procedure.

Line 94-97: please delete this sentence. If you didn’t modify procedure, it is not necessary to explain kit’s instructions.

Line 107: which “closely related non-target species” did you considered? The table 2 reports other species, but they are not closely related to Thunnus albacares

Line 135: you used a ladder from 2000 bp to 50 bp, please specify the bp for your expected amplicons

Line 136: please, add a reference for “Direct Detection by MSwab” method

Line 152: you tested the specificity of the LAMP assay on your 18 samples. Did you use an internal control (COI, 16S, ecc) to be sure that your samples were exactly Thunnus albacares? Since some tuna species (T. obesus, albacares, alalunga, ecc.) may be different just for point mutation and produce the same Melting T°.

Line 152: you used 18 samples of the target species Thunnus albacares, plase specify it

Line 153: The “other fish species (non-target species)” listed in table 2 are not 18, you have 18 samples but only 5 “other” species.

Line 160: please replace “tenfold” with “ten-fold”.

Line 176: please replace “tenfold” with “ten-fold”.

Line 191: please replace “meat products” with “food products”

Line 197: please replace the sentence with “plants (25, 26), animals (20, 27-30), bacteria (31, 32), and herbal medicine.”

Line 201: please add the spot at the end of the sentence.

Line 206: please add the space after “LAMP”.

Line 208: please replace “species specific primers” with “species-specific primers”.

Line 240: please delete the repetition of the word “assay”.

Fig 4: Please, specify the bp for your amplicons.

Reviewer #2: 1.This work needs a deep review of English throughout the document.

2.In this study, the authors describe an assay for yellowfin tuna with a lower limit of detection of 540 fg/μl and good specificity as well as providing a rapid DNA extraction method.

3.In my opinion, the authors needed to benchmark a recognized assay for 36 samples, and fluorescent quantitative PCR（qPCR） was a good choice.

4. LAMP as a common method for isothermal amplification needs to be compared with other isothermal amplification methods such as RPA, RCA, CPA, SDA, and HDA, and the authors need to add this section in the Introduction and Discussion section.

5.The authors mentioned that the detection target sequences were compared, and the results of this section need to be presented as supplementary material.

6.Changes need to be made to Figure 4, and I think the left Marker is best deleted.

Reviewer #3: A rapid method for yellowfin tuna (Thunnus albacares) authentication by LAMP was developed in the present work. Generally, I would say it is an interesting work, and the potential of LAMP in seafood species identification has been verified. However, there are lots of problems with the present one:

1) the results should be carefully described,

2) the figures and tables should be detailedly explained

3) the comparison between traditional method and rapid extraction method should be described and explained

4) besides gel electrophoresis, visual detection showed more advantages, which are not used in the present work.

6. PLOS authors have the option to publish the peer review history of their article (what does this mean?). If published, this will include your full peer review and any attached files.

Reviewer #1: No

Reviewer #2: No

Reviewer #3: No

---

## [Author Response · Author response to Decision Letter 0]

15 Aug 2022

The authors would like to thank the reviewers for the constructive comments which helped to improve this manuscript. In the rebuttal letter the authors refer to all comments and hope to have addressed all aspects adequately.

---

## [Decision Letter · Decision Letter 1]

19 Sep 2022

Development of loop-mediated isothermal amplification (LAMP) assay for rapid and direct screening of yellowfin tuna (Thunnus albacares) in commercial fish products

PONE-D-22-09199R1

Dear Dr. Abdulmawjood,

We’re pleased to inform you that your manuscript has been judged scientifically suitable for publication and will be formally accepted for publication once it meets all outstanding technical requirements.

Kind regards,

Aldo Corriero, Ph.D.

Academic Editor

PLOS ONE

Additional Editor Comments (optional):

Reviewers' comments:

Reviewer's Responses to Questions

**Comments to the Author**

1. If the authors have adequately addressed your comments raised in a previous round of review and you feel that this manuscript is now acceptable for publication, you may indicate that here to bypass the “Comments to the Author” section, enter your conflict of interest statement in the “Confidential to Editor” section, and submit your "Accept" recommendation.

Reviewer #2: All comments have been addressed

Reviewer #3: (No Response)

2. Is the manuscript technically sound, and do the data support the conclusions?

Reviewer #2: Yes

Reviewer #3: No

3. Has the statistical analysis been performed appropriately and rigorously? 

Reviewer #2: Yes

Reviewer #3: No

4. Have the authors made all data underlying the findings in their manuscript fully available?

Reviewer #2: Yes

Reviewer #3: No

5. Is the manuscript presented in an intelligible fashion and written in standard English?

Reviewer #2: Yes

Reviewer #3: No

6. Review Comments to the Author

Reviewer #2: I agree to this manuscript for publication on Plos one，I agree to this manuscript for publication on Plos one.

Reviewer #3: The Author response is missing, and the revised manuscript still has lots of errors.

The results section and the figures/tables, as suggested in my previous comment, should be carefully described.

The present one is not acceptable.

7. PLOS authors have the option to publish the peer review history of their article (what does this mean?). If published, this will include your full peer review and any attached files.

Reviewer #2: No

Reviewer #3: No

---

## [Editor Report · Acceptance letter]

20 Sep 2022

PONE-D-22-09199R1 

Development of loop-mediated isothermal amplification (LAMP) assay for rapid and direct screening of yellowfin tuna (*Thunnus albacares*) in commercial fish products 

Dear Dr. Abdulmawjood:

I'm pleased to inform you that your manuscript has been deemed suitable for publication in PLOS ONE. Congratulations! Your manuscript is now with our production department. 

Kind regards, 

on behalf of

Dr. Aldo Corriero 

Academic Editor

PLOS ONE